# Effect of Sulfonic Groups Concentration on IEC Properties in New Fluorinated Copolyamides

**DOI:** 10.3390/polym11071169

**Published:** 2019-07-09

**Authors:** Ramón de Jesús Pali-Casanova, Marcial Alfredo Yam-Cervantes, José del Carmen Zavala-Loría., María Isabel Loría-Bastarrachea, Manuel de Jesús Aguilar-Vega, Luis Alonso Dzul-López, María Luisa Sámano-Celorio, Jorge Crespo-Álvarez, Eduardo García-Villena, Pablo Agudo-Toyos, Francisco Méndez-Martínez

**Affiliations:** 1Department of the Doctorate in Industrial Engineering, Universidad Internacional Iberoamericana A.C. Calle 15 No. 36 x 10 y 12. IMI III, Campeche C.P. 24560, Campeche, Mexico; 2Centro de Investigación Científica de Yucatán, Materials Unit, Calle 43 No. 130 x 32 y 34 Chuburná de Hidalgo, Mérida C.P. 97205, Yucatán, Mexico; 3Centro de Investigación y Tecnología Industrial de Cantabria (CITICAN), Calle Isabel Torres No. 21, 39011 Santander, Spain; 4Department of the Doctorate in Industrial Engineering, Universidad Europea del Atlántico, Calle Isabel Torres No. 21, 39011 Santander, Spain; 5Facultad de Ingeniería de la Universidad Autónoma del Carmen. Calle 56 No. 4 Esq. Avenida Concordia Col. Benito Juárez, Cd. del Carmen C.P. 24180, Campeche, Mexico

**Keywords:** polyamides, sulfonation, ion exchange capacity, proton conductivity

## Abstract

Seven aromatic polyamides and copolyamides were synthesized from two different aromatic diamines: 4,4′-(Hexafluoroisopropylidene)bis(p-phenyleneoxy)dianiline (HFDA) and 2,4-Aminobenzenesulfonic acid (DABS). The synthesis was carried out by polycondensation using isophthaloyl dichloride (1SO). The effect of an increasing molar concentration of the sulfonated groups, from DABS, in the copolymer properties was evaluated. Inherent viscosity tests were carried out to estimate molecular weights. Mechanical tests were carried out under tension, maximum strength (σ_max_), Young’s modulus (*E*), and elongation at break (ε_max_) to determine their mechanical properties. Tests for water sorption and ion exchange capacity (IEC) were carried out. Proton conductivity was measured using electrochemical impedance spectroscopy (EIS). The results indicate that as the degree of sulfonation increase, the greater the proton conductivity. The results obtained showed conductivity values lower than the commercial membrane Nafion 115 of 0.0065 S cm^−1^. The membrane from copolyamide HFDA/DABS/1S0-70/30 with 30 mol DABS obtained the best IEC, with a value of 0.747 mmol g^−1^ that resulted in a conductivity of 2.7018 × 10^−4^ S cm^−1^, lower than the data reported for the commercial membrane Nafion 115. According to the results obtained, we can suggest that further developments increasing IEC will render membranes based on aromatic polyamides that are suitable for their use in PEM fuel cells.

## 1. Introduction

Proton exchange membranes used for fuel cells (PEMFCs) are systems that have been extensively developed in recent years for terrestrial applications, especially by the automotive industry, which has announced its interest in the development of all electric autonomous cars. Fuel cells have the following advantages: They present high efficiency, low contaminant emissions, zero noise, and a clean energy source. Several types of fuel cells have been developed, including: Alkaline Fuel Cells (AFC), Direct Methanol Fuel Cells (DMFC), Phosphoric Acid Fuel Cells (PAFC), Molten Carbon Fuel Cells (MCFC), Solid Oxides Fuel Cells (SOFC), and Proton Exchange Membrane Fuel Cells (PEMFC) [1].

In the case of PEMFCs′ the main component of the proton exchange fuel cell is the membrane electrolyte assembly, MEA, which under constant conditions generates 0.4 to 1 V. Under a bipolar array (stack series), it is possible to increase the voltage, making it attractive for several applications [1,2,3]. However, PEMFCs are expensive and unattractive for terrestrial applications because of several drawbacks. The most important is related to “cost”, since electrodes in the PEMFCs are based on platinum (Pt), as a catalyst for hydrogen (H_2_) decomposition. The second order of “cost” priority corresponds to the polymeric ionic membrane used, a perfluorinated sulfonated material, from which the benchmark is Nafion^®^, which has a high cost and as it is difficult to dispose of, thus becomes a contaminant [1,2,3]. In order to decrease costs, there have been efforts to develop a non-Pt catalyst that will perform the H_2_ separation, as well as the production of lower-cost alternative PEMFC membranes, with lower manufacturing costs, to produce PEMFC devices competitive in today′s energy market [1,4].

One of the characteristics of a perfluorinated ion exchange membrane is its ability to transport protons from the anode to the cathode in the PEMFC, restricting the passage of electrons. The membrane should show good thermal and chemical stability, as well as mechanical properties. It also has to present low permeability for fuels and oxidants [2,5]. Among the membranes that meet these requirements, Nafion^®^ 115, developed by Dupont, is the benchmark due to its balance of properties [1,2,3,4,5].

Although Nafion^®^ 115 performance is superior to other commercial polymeric membranes, its high cost, low conductivity at temperatures above 80 °C, and humidity loss when working at high temperatures contributes to the search for developing alternative polymers to improve these deficiencies [1,3,4,5].

A promising route has been tested where research focuses on aromatic polymers with sulfonic groups (SO_3_H). They provide hydrophilic characteristics similar to commercial polymers. Sulfonated aromatic polymers usually have high thermal and chemical stability that makes them an alternative for PEMFC membranes production. The aromatic character of these polymers makes them highly resistant to degradation, which favors proton conduction. However, the conditions to which these membranes are subjected in the cells are very different from what these polymeric membranes were originally designed for [6,7,8,9].

During their performance, membranes are generally exposed to oxidation through the cathode, and reduction through the anode. Both occur in the presence of catalytic activity and are accompanied by membrane swelling due to the hydrophilic nature of the sulfonic groups (SO_3_H), which contributes to making these membranes capable of absorbing water or solvent mixtures such as methanol/water. In addition, the polymer′s sulfonation degree is related to its water sorption capacity. As the sulfonation degree increases, the sorption of water and/or polar solvents increases. On the other hand, the mechanical properties are reduced due to the presence of the polar solvent absorbed in the polymer, which leads to a decrease in performance within the cell [6,7,8,9]. Some of the polymers that have been reported, such as sulfonated polyether ether ketone, S-PEEK [10,11], sulfonated polyether ether ketone ketone, S-PEEKK [12,13], sulfonated polyether sulfone, S-PESU [14], sulfonated polysulfone, S-PSU [14,15], and sulfonated arylene ethers, S-PAE [16,17], present a performance similar to Nafion^®^ 115. However, the degradation of the membrane occurs faster in aromatic polymers than in perfluorinated ones. A study determined that after 228 service hours, the aromatic polymers started experiencing degradation, whereas the perfluorinated polymers sustained their activity up to 400 h [2,3,5,9] under the same conditions.

There has been some efforts carried out lately to find sulfonated polymer substitutes for perfluorinated ionic polymers for fuel cells [8,18]. The synthesis and characterization of new sulfonated polyamides and their membranes will allow us to explore the possibility of finding out if their properties could match those of Nafion^®^ membranes for PEMFC application. It will be also important to estimate the effect that an increase in sulfonic groups concentration, SO_3_H, with the preparation of aromatic copolyamides, has on the properties and stability of the membranes for fuel cell performance [9].

This work reports the synthesis of one aromatic polyamide and 6 sulfonated copolyamides obtained from 4,4′-(Hexafluoroisopropylidene)bis(p-phenyleneoxy)dianiline, HFDA, 2,5-Diamino benzenesulfonic acid, DABS, and isophthaloyl dichloride, ISO, with increasing DABS concentration maintaining a minimum degree of fluorine groups in the main chain structure. The characterization of a these new copolyamides (HFDA, DABS, ISO) will be carried out to determine their mechanical properties, water absorption capacity, ion exchange capacity, and proton conductivity to confirm that they show properties that are suitable for application in PEMFCs [7,9,19,20].

## 2. Materials

4,4′-(Hexafluoroisopropylidene)bis(p-phenyleneoxy)dianiline, HFDA, (97%), 2,5-Diamino benzenesulfonic acid, DABS, (97%), isophthaloyl dichloride, ISO, (99%), triethylamine, TEA, (99%), 1,1,2,2-Tetrachloroethane, TCE, (98%), *N-N*-Dimethylacetamide, DMAC, phenolphthalein (99%), hydrochloric acid, HCl, sodium chloride, NaCl. All the reagents used were purchased from Aldrich Chemical Inc. (St. Louis, MO, USA), with the exception of HCl which was obtained from JT Baker (Phillipsburg, NJ, USA).

## 3. Methodology

### 3.1. Copolyamide Synthesis

#### 3.1.1. ISO Purification Method

In a 120 mL beaker, 2 g isophthaloyl dichloride, ISO, and 35 mL of petroleum ether were added. The solution was stirred until ISO was completely dissolved at room temperature, then the solution was filtered with Whatman paper no. 40. After this point, the solution was poured and recrystallized in an ice bath for 12 h. The final product was in the form of fine and translucent crystals [16]. The triethylamine was purified by distillation at atmospheric pressure. Distillation was carried out in a round-bottom flask equipped with a micro-condenser; 4 mL of TEA was added and brought to 89 °C (TEA boiling point) and recovered as shown in Figure 1 [16,21].

#### 3.1.2. Synthesis of Sulfonated Copolyamides

The copolyamides synthesis bearing different concentrations of DABS and HFDA diamines (DABS (*Y* = 5, 10, 20, 30, 40, and 50 mol%)) (Figure 1) was carried out at high temperature using isophthaloyl dichloride. A typical copolymerization reaction for HFDS/DABS/ISO-90/10 is as follows: In a three-neck 250 mL round-bottom flask equipped with a magnetic stirrer, 2 mmol (0.4 g ISO), 2 mmol diamines (*X* = HFDA 90 mol% and *Y* = DABS 10 mol% equivalent to 0.9620 and 0.0388 g, respectively) in 6 mL of 1,1,2,2-tetracholoroethane, TCE, were added. The solution was stirred until it completely dissolved at room temperature. To initiate the reaction, 1.12 mL of TEA was added dropwise, stirring constantly. The reaction solution was kept at 90 °C for 9 h under nitrogen atmosphere. After this time the reaction was cooled down to 70 °C and precipitated in 500 mL of methanol. The precipitated copolymer was washed several times with hot water. Finally, the polymer obtained was dried in a convection oven at 120 °C for 24 h [16,21]. All other sulfonated copolyamides were prepared using the same procedure with the appropriate variation in concentration of the diamines DABS and HFDA. Inherent viscosity (η_inh_) for HFDA/ISO polyamide and HFDA/DABS/ISO sulfonated copolyamides was determined in a No. 50 Cannon Ubbelohde viscometer (Tokyo, Japan) with a concentration of 0.5 g dl^−1^ of polymer solution in *N*, *N*-Dimethyl acetamide, DMAC, at 30 °C.

### 3.2. Film Preparation

Films of HFDA/ISO and sulfonated copolyamides, HFDA/DABS/ISO, were cast using DMAC by the solution evaporation method. A total of 0.5 g of polyamide or copolyamide were dissolved in 5 mL of DMAC for 24 h. The solution obtained was then filtered and poured into a round aluminum mold with the following dimensions: Diameter 8 cm and height 2 cm, and placed on a heating plate at a temperature of 80 °C for 24 h. The residual solvent was removed by drying at 160 °C for a period of 24 h [16,21]. To be sure that there was not residual solvent, thermogravimetric analysis was performed on the membranes.

### 3.3. Characterization

#### 3.3.1. Mechanical Properties

In order to study the mechanical property effects on the sulfonated structure, tensile stress tests were carried out using polyamide and sulfonated copolyamide films cut to the following dimensions: 12.5 mm long × 5 mm wide, with a thickness measured for each one that was between 0.037 and 0.052 mm. The membranes were previously dried at 100 °C for 24 h to eliminate possible absorbed water. The tests were performed using a Minimat tensile tester (Rheometrics Inc., Piscataway, NJ, USA) with a 100 N load cell and a 2 mm/min head displacement, at room temperature, according to procedures described in the standard method ASTM D-882. For the measurements, 5 samples per membrane were prepared and from each stress-strain curves the Young’s Modulus, *E*, maximum stress (σ_max_), and elongation at break (ε_max_) were calculated [21,22,23].

#### 3.3.2. Water Absorption Capacity (WAC)

The water absorption capacity was measured by gravimetric analysis. For the test, 1cm × 1 cm samples of copolyamide membranes were cut and vacuum dried at 120 °C for 24 h. After drying the membranes were weighed (*m*_dry_) and immersed in deionized water for 48 h at different temperatures (25, 45, 65, and 75 °C). After each process the membranes were weighed to determine the wet weight (*m*_wet_). Lastly, the amount of water retained was calculated as the percentage of membrane weight gain due to water absorbed using Equation (1) [16,21,23].
(1)$$WAC=\frac{m_{wet}-m_{dry}}{m_{dry}}\times 100$$

#### 3.3.3. Ion Exchange Capacity (IEC)

The Ion Exchange capacity, IEC, was determined using a titration method. In this test, 1.5 cm × 0.5 cm membrane samples were dried at 120 °C for 24 h; then the membranes were weighed, *m_dry_*, and immersed in a 1 M hydrochloric acid (HCl, 1 M) solution for 24 h to obtain membranes in their acid form; in the next step, they were washed in deionized water until neutral pH. The membranes were immersed in 1 M NaCl solution in order to exchange the H^+^ with the Na^+^ ions. Using, as an indicator, a solution of phenolphthalein/ethyl alcohol at 1 %, the NaCl solution was titrated with 0.01 M NaOH solution until it changed to a tenuous pink color. By knowing the volume of consumption, the *IEC* was determined for each sample in triplicate using Equation (2) [16,21].
(2)$$IEC=\frac{C_{NaOH}V_{NaOH}}{m_{dry}}$$

#### 3.3.4. Proton Conductivity Determination

The membranes form HFDA/ISO and sulfonated HFDA/DABS/ISO were cut with a 13 mm diameter and then immersed in deionized water for 96 h, before being removed from the water and immersed in a solution of 1M HCl for 12 h and finally thoroughly washed with water until neutral pH. Proton conductivity was measured by electrochemical impedance spectroscopy (EIS). The measurements were performed using a PGSTAT12/30/302 Autolab Potentiostat-Galvanostat Autolab PGSTAT 302 Eco Chemie (Metrohm Autolab V.V, Utrecht, The Netherlands) by means of an EIS module over a frequency range of 10^6^ to 1 Hz, with a current amplitude of 10 V. The films were placed between two stainless steel alloy 20 electrodes (13 mm diameter). The resistance value associated with the membrane conductivity was determined from the Nyquist diagram taking the high frequency intercept of the impedance with the real axis. The proton conductivity, σ (S cm^-1^) of each membrane was calculated according to the following equation [16,23].
(3)$$\sigma =\frac{f_{t}}{RA}$$
where *f*_t_ is the film thickness (cm), *R* is the membrane calculated resistance (Ω), and *A* is the membrane area (cm^2^).

## 4. Results

A total of six sulfonated copolyamides with different degrees of sulfonation (5 to 50 mol%) were obtained by the polycondensation reaction using two different diamines, one fluorinated and the other sulfonated reacting with diacid chloride. Table 1 presents the base polyamide HFDA/ISO, with non-sulfonic groups present, and the six random copolymers HFDA/DABS/ISO-*X*/*Y* where *X* and *Y* indicate the moles of non-sulfonated and sulfonated diamine.

As seen from Table 1 results, inherent viscosity values, η_inh_, an indication of the copolymer molecular weight, decreased as DABS increased in the copolymer, the lowest being for HFDA/DABS/ISO-50/50, and the results are comparable with the data reported in the literature for other copolyamides [24,25]. This result indicates that the presence of the sulfonated group in the reaction media may inhibit the copolymer formation by reacting with some of the isophthaloyl chloride, decreasing the molecular weight.

Films from sulfonated polymers were obtained only for those copolyamides with DABS concentration lower that 30 mol% that precipitate in the form of fibers. They presented a decrease in mechanical properties, as the concentration of DABS increased in the copolymer. This behavior is similar to the one reported in literature for other sulfonated copolyamides [19,26,27]. The mechanical properties of the copolymer films obtained were maximum tensile stress (σ*_max_*), elongation at break (ɛ*_max_*), and Young′s modulus (*Е*), which are reported in Table 2. The results indicate a decrease in mechanical properties with increasing presence of DABS in the sulfonated copolyamide, which may be attributed to the lowering of the molecular weight since they closely follow the tendency of inherent viscosity decrease shown in Table 1.

### 4.1. Water Absorption Capacity (WAC)

HFDA/DABS/ISO membrane films with higher degree of sulfonation showed increased water absorption as DABS moiety content increased in the copolyamide. The water absorption capacity increased also as the temperature increased between 25 and 75 °C; the concentration of sulfonated diamine (DABS) also increased. This result is attributed to the fact that the copolymer presents increasing hydrophilicity, with the increased concentration of sulfonic groups (DABS) [28,29,30]. It is also seen that as temperature increased the water uptake increased closely following a linear behavior with temperature (see Figure 2).

The results are interesting because the WAC absorption of the HFA/DABS/ISO-95/5 to HFDA/DABS/ISO-70/30, compared to the one of Nafion^®^ 115 membrane, was similar or larger at all temperatures tested. The results indicate that even though the mobility of the copolyamides chains is lower than the ones of Nafion^®^, which is a rubbery polymer, the increased presence of sulfonated groups in the rigid copolyamides allowed a greater water absorption than Nafion^®^, as reported in Table 3 [29,30].

Membranes from sulfonated copolyamides with 40 mol% and 50 mol% were not characterized because they were very brittle and broke during the absorption test.

### 4.2. Ion Exchange Capacity (IEC)

The IEC of the HFDA/ISO and HFDA/DABS/ISO copolymers synthesized in this work are compared to commercial Nafion 115 membrane in Figure 3. The random copolymer that had only 5 mol% DABS showed a very low increase in ion exchange capacity (IEC), but from 10 to 30 mol% DABS the IEC increased almost linearly between 0.48 to 0.74 mmol g^−1^. This is still below the value reported for Nafion 115 of 1.28 mmol g^−1^ [8], even though the WAC was larger. The difference seems to be related to the connectivity of the sulfonated domains that have a random distribution in HFDA/DABS/ISO copolymers while they are interconnected in Nafion^®^ 115.

In Table 4 it can be observed that as the concentration of the sulfonic groups increases, IEC also increases, which is consistent with other works reported in the literature [8,20].

### 4.3. Proton Conductivity 

Figure 4a,b shows a typical Z′′ vs Z′ Nyquist plot for the electrochemical impedance spectroscopy (EIS) tests for samples HFDA/ISO and a sulfonated copolymer HFDA/DABS/ISO-90/10. The proton conductivity obtained after a non-linear adjustment was made using the circular function to obtain the characteristic resistance of the copolymer membrane, *R*_o_, obtained by extrapolation of the adjustment curve to the frequency zone (Z′′ →0). For all the copolymers, using Origin 8 software, the intersection with the real axis was adjusted to calculate resistance (*R*_o_). Based on these *R*_o_, using Equation 3, the conductivity for HFDA/ISO and each HFDA/DABS/ISO sulfonated copolymer was calculated 5b. As in the case of IEC values, the conductivity of the copolymers increased with increasing DABS concentration. However, the proton conductivity increased with the copolyamide with the largest concentration of sulfonated groups that was possible to test, HFDA/DABS/ISO-70/30, is one order of magnitude below the one obtained for Nafion^®^ 115 under the same testing conditions. Thus, while polyamides sulfonation seems a promising route for obtaining membranes to be used in PEMFCs, there is a need to improve their IEC and conductivity by carefully tailoring their structure to improve the connectivity between sulfonated domains.

As can be seen in Figure 5, an increase of 20 to 30 mol% of sulfonic groups increased the membrane conductivity by one order of magnitude. Comparing these results with commercial membranes, such as Nafion^®^ 115, we can expect that, with a proper balance between structure and sulfonation degree, the as-synthesized membranes could become good candidates to be developed for use as PEM membranes in fuel cells, as well as potential applications for automobile batteries and photovoltaic cells, among other applications [30,31].

## 5. Conclusions 

The synthesis of six new aromatic sulfonated copolyamides (HFDA/DABS/ISO-*X*/*Y*) was carried out. Inherent viscosity tests indicate that an increasing concentration of the sulfonated diamine in the copolyamide structure decreases the molecular weight of the sulfonated copolyamide. Mechanical properties under tension, such as maximum stress at break (σ_max_), Young modulus at tension (*E*), and elongation at break (ε*_max_*), were measured and they decrease as the inherent viscosity of the copolymer decreases. Through tests of water absorption capacity and the ion exchange capacity of the membranes (IEC), it was possible to determine that the water absorption capacity (WAC) and ion exchange capacity (IEC) increased with increasing concentration of sulfonated groups in the copolymer. The proton conductivity showed that, as the degree of sulfonation is larger, the conductivity of the HFDA/DABS/ISO copolymer enhances. The results obtained showed conductivity values that are lower than the commercial membrane Nafion^®^ 115. The best performing *IEC* was for the HFDA/DAB5/1S0-70/30 membrane, with a value of 0.74 mmol g^−1^, which resulted in a conductivity of 2.7 × 10^−4^ S cm^−1^, which is lower than the data reported for the commercial Nafion 115 membrane. According to the results obtained, we can suggest that these membranes based on aromatic polyamides have to be further developed to make them suitable for use in PEM fuel cells.

## Figures and Tables

**Figure 1 polymers-11-01169-f001:**
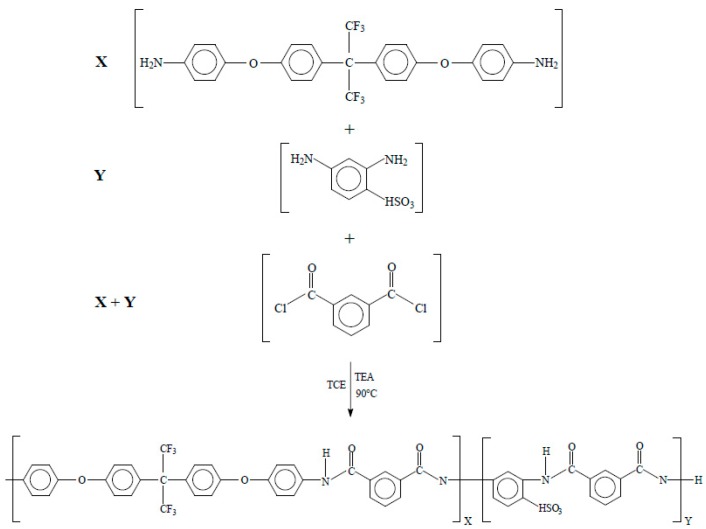
Schematic of polycondensation reaction for synthesis of HFDA/ISO polyamide (*X* = 100 mol %, *Y* = 0 mol%) and sulfonated copolyamides HFDA/DABS/ISO (*X* = 95, 90, 80, 70, 60, and 50 mol %).

**Figure 2 polymers-11-01169-f002:**
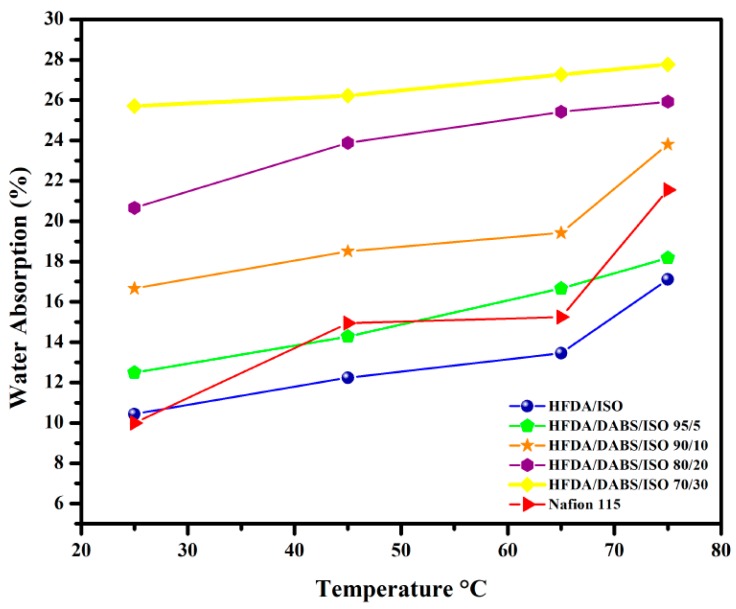
Water absorption capacity at different temperatures for HFDA/ISO and HFDA/DABS/ISO copolyamides.

**Figure 3 polymers-11-01169-f003:**
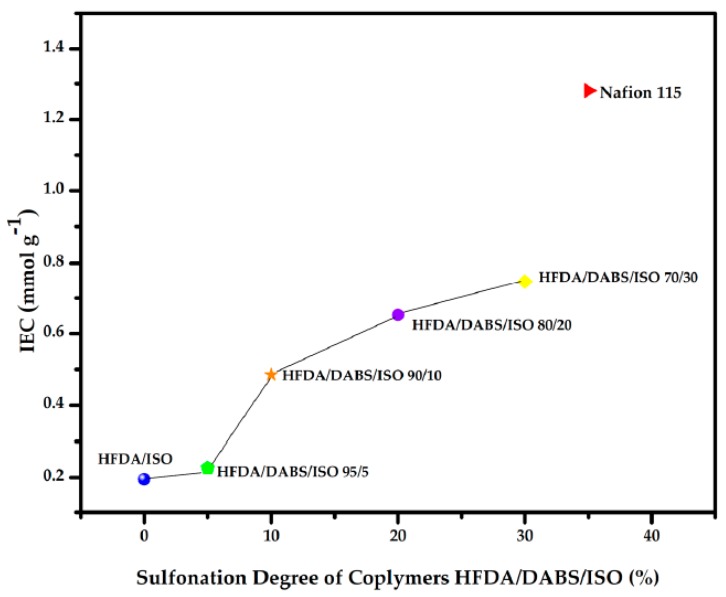
IEC of the membranes HFDA/ISO and HFDA/DABS/ISO based on the degree of sulfonation.

**Figure 4 polymers-11-01169-f004:**
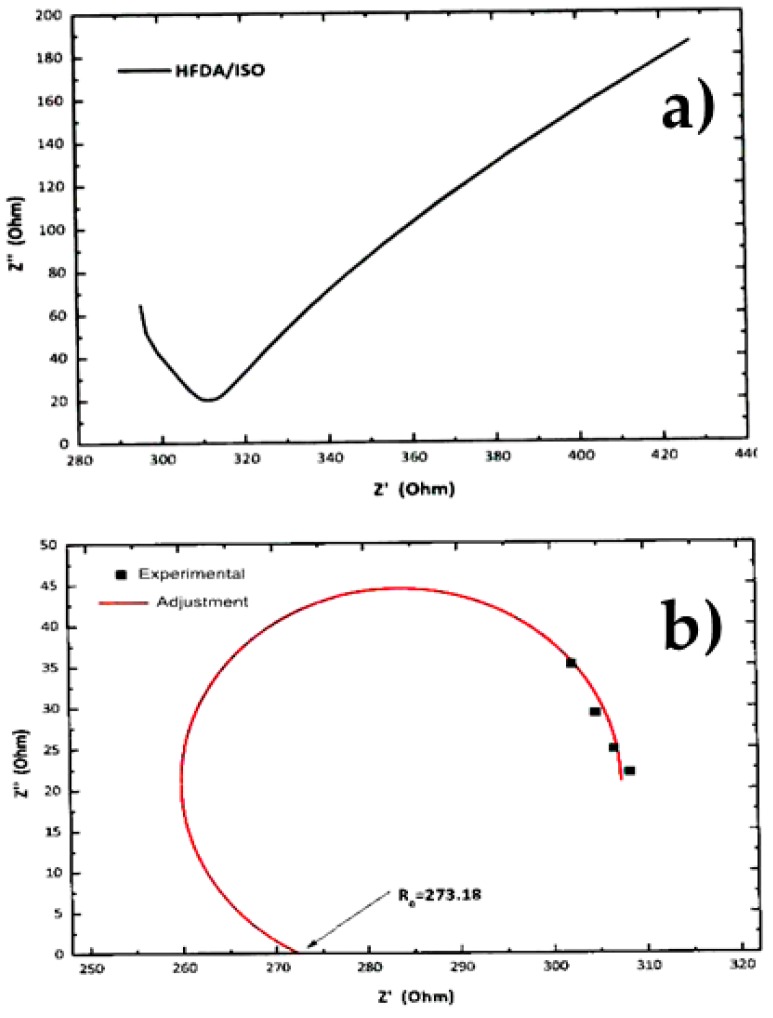
Characteristic EIS result and non-linear adjustment of (**a**) EIS curve for non-sulfonated HFDA/ISO, (**b**) adjustment for HFDA/DABS/ISO-90/10.

**Figure 5 polymers-11-01169-f005:**
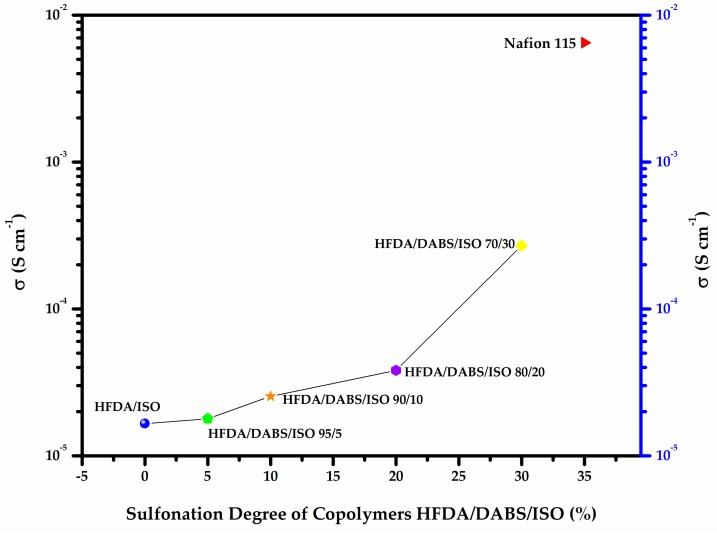
Proton conductivity of HFDA/ISO and *HFDA/DABS/ISO* membranes as a function of sulfonation degree compared to Nafion^®^ 115.

**Table 1 polymers-11-01169-t001:** Homopolymer and random copolymer prepared by polycondensation.

Polymer	Product Obtained	Amount (g)	Film	η_inh_ (dL/g)
HFDA/ISO	FIBERS	1.12	✓	0.5482
HFDA/DABS/ISO-95/5	FIBERS	0.96	✓	0.4487
HFDA/DABS/ISO-90/10	FIBERS	0.80	✓	0.483
HFDA/DABS/ISO-80/20	FIBERS	0.72	✓	0.3659
HFDA/DABS/ISO-70/30	NON-DEFINED FIBERS	0.66	✓	0.3163
HFDA/DABS/ISO-60/40	SEMI-POWDER	0.60	X	0.3066
HFDA/DABS/ISO-50/50	POWDER	0.54	X	0.2581

**Table 2 polymers-11-01169-t002:** Mechanical properties under tension for HFDA/ISO and HFDA/DABS/ISO copolymers.

Polymer	Thickness (mm)	σ*_max_* (MPa)	ɛ*_max_* (%)	*Е* (GPa)
HFDA/ISO	0.047	60.36	11.11	1.069
HFDA/DABS/ISO 95/5	0.037	53.15	7.45	1.002
HFDA/DABS/ISO 90/10	0.041	42.24	5.51	0.954
HFDA/DABS/ISO 80/20	0.043	50.97	5.85	1.061
HFDA/DABS/ISO 70/30	0.052	34.4	2.34	0.788
HFDA/DABS/ISO 60/40	-	-	-	-
HFDA/DABS/ISO 50/50	-	-	-	-

**Table 3 polymers-11-01169-t003:** Water absorption capacity (%) of HFDA/ISO and HFDA/DABS/ISO sulfonated copolyamides at different temperatures.

Polymer	25 °C	45 °C	65 °C	75 °C
HFDA/ISO	10.44	12.24	13.46	17.11
HFDA/DABS/ISO 95/5	12.5	14.28	16.66	18.18
HFDA/DABS/ISO 90/10	16.66	18.51	19.42	23.80
HFDA/DABS/ISO 80/20	20.96	23.88	25.42	25.92
HFDA/DABS/ISO 70/30	25.71	26.22	27.27	27.77
HFDA/DABS/ISO 60/40	-	-	-	-
HFDA/DABS/ISO 50/50	-	-	-	-
Nafion 115	10	14.95	15.25	21.55

**Table 4 polymers-11-01169-t004:** IEC results of copolyamides based on HFDA/ISO and HFDA/DABS/ISO copolymers performed at room temperature.

Polymer	0.01 M NaOH Added (mL)	Membrane dry Weight(g)	IEC (mmol/g)
HFDA/ISO	0.275	0.0141	0.1954
HFDA/DABS/ISO 95/5	0.360	0.0158	0.2274
HFDA/DABS/ISO 90/10	0.775	0.0160	0.4847
HFDA/DABS/ISO 80/20	1.175	0.0180	0.6527
HFDA/DABS/ISO 70/30	0.890	0.0119	0.7478
HFDA/DABS/ISO 60/40	-	-	-
HFDA/DABS/ISO 50/50	-	-	-

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
