# Peer review of "Effect of Sulfonic Groups Concentration on IEC Properties in New Fluorinated Copolyamides"

_polymers, 2019, doi:10.3390/polym11071169_

Reviewer 1 Report

- make sure all abbreviations are at least once writte full, e.g. what does GDE mean in the introduction? what does TCE mean in the experimental part?

- elaborate why isophtalic acid is used and not terephtalic acid

- In the experimental part you write that the polycondensation was initiated by addition of TCE. I suppose TCE is tetrachloro ethanol, a solvent. How can polycondensation be initiated by solvents?

- In table 1 you mention inherent viscosity values, but in the experimental part it is not described how those values were measured. Please add

- Table 1, HFDA/ISO: should't fibras be fiber?

- film preparation: how sure are you that all DMAc is removed?

- In some tables you mention the Nafion 115 reference, in others not. In the conclusions you try to compare your membranes compared to Nafion 115, but I miss their properties in most tables of the article.

- Table 2: HFDA/DABS/ISO 90/10 Shouldn't Emax be 0.954 intead of 0.0954?

- Fig 4: why do you have this solid line?

- Table 3. add that the water absorption capacity is expressed in %

- Fig 6, bottom: I think you do not have enough data points to make this extrapolation. Can you generate more data points?

- Table 4: why do you add different amounts of NaOH for each sample?

- Conclusions: you mention Nysquist diagrams, but this terminology was not used in the results and discussion. Add.

Author Response

Please see the attachment in the box 

Reviewer 2 Report

Most of the used acronyms have not been explained, for example: IEC (in the title); HFDA/DABS/1S0-70/30 ;  PEMFC, GDE or PEM (in the abstract and in the introductory part) 

Some formulation are used incorrectly for example:  

 maximum resistance, 

 elongation at rupture, Maximum Effort, 

The level of English has to be improved. There are some parts of the manuscript which are unacceptable:

“The results obtained…”, „During their performance, membranes are generally exposed to oxidation conditions”;  “They provide hydrophilic and similar characteristics to commercial polymers”;

“In addition, the polymer’s degree of sulfonation guides its solubility capacity”

The section dedicated to the “Preparation and Synthesis of sulfonated copolyamides” is unclear. Authors mentioned that components ( ) were dissolved in TCE, however in the next line (line 130) we can find information that “To initiate the reaction, 1.12 ml of TCE were added by dripping”. Moreover in the text we can read that reaction of HFDA-DABS was performed while in the picture the reaction for copolyamide HFDA/DABS/ISO has been depicted. 

One of the sentences indicates that “2 mmol diamines (HFDA 90 % and DABS 10%) in 6 ml 129 TCE were added” (lines 129-130). This statement is grossly inaccurate since, first of all, - DABS is not a diamine, second of all,  there is no clear indication as to the exact amounts of components used in the experiment.

- One part of manuscript is written in the past tense another one in the present tense, In particular the copolymer and film formation procedures need to be amended in terms of the grammar tense.

- Mechanical properties should be determined in accordance with a  relevant standard. 

- The conditions of “Inherent viscosity” analysis have not described.

- There is no information related to the homopolymer HFDA/ISO formation. Bearing in mind that HFDA as well as ISO have two amine groups I would like to see the proposed reaction schematic.

- In the tree-component copolymers such as HFDA/DABS/ISO-95/5 there are only values which apparently describe the amount of substrates . The designations of the samples should comprise values pertaining to all monomers that form the copolymer.

- Taking into account mechanical properties the stress-strain curves should have been presented.

In the main text and tables we can find that 6 sulphonated materials were obtained, however in the conclusion the authors claim that “seven new aromatic sulphonated polyamides” were synthesized.

The manuscript seems to be in many ways a work in progress and incomplete. The procedures of syntheses, designations of samples, reaction schematics and inadequate  English level make this work unsuitable for publication.

Author Response

Please see the attachment in the box

Round  2

Reviewer 1 Report

accepted in revised form

Author Response

Please see the attachment in the box 

Thank you so much.
